# Dietary Intervention in Pregnant Women with Gestational Diabetes; Protocol for the DiGest Randomised Controlled Trial

**DOI:** 10.3390/nu12041165

**Published:** 2020-04-22

**Authors:** Laura C. Kusinski, Helen R. Murphy, Emanuella De Lucia Rolfe, Kirsten L. Rennie, Linda M. Oude Griep, Deborah Hughes, Roy Taylor, Claire L. Meek

**Affiliations:** 1Institute of Metabolic Science, University of Cambridge, Cambridge CB2 0QQ, UK; lck34@medschl.cam.ac.uk (L.C.K.); djh251@medschl.cam.ac.uk (D.H.); 2Cambridge Universities NHS Foundation Trust, Cambridge CB2 0QQ, UK; helen.murphy@uea.ac.uk; 3Norwich Medical School, University of East Anglia, Norwich NR4 7UQ, UK; 4NIHR Cambridge Biomedical Research Centre—Diet, Anthropometry and Physical Activity Group, MRC Epidemiology Unit, Institute of Metabolic Science, University of Cambridge, Cambridge CB2 0QQ, UK; emanuella.de-lucia-rolfe@mrc-epid.cam.ac.uk (E.D.L.R.); Kirsten.Rennie@mrc-epid.cam.ac.uk (K.L.R.); Linda.OudeGriep@mrc-epid.cam.ac.uk (L.M.O.G.); 5Institute of Cellular Medicine, University of Newcastle, Cambridge NE4 5PL, UK; roy.taylor@ncl.ac.uk

**Keywords:** gestational diabetes, pregnancy, study protocol, randomised controlled trial, large-for-gestational age, diet, dietary intervention, maternal or gestational weight gain, continuous glucose monitoring (CGM), neonatal outcomes, neonatal hypoglycaemia, neonatal growth, maternal obesity

## Abstract

Gestational diabetes mellitus (GDM) annually affects 35,000 pregnancies in the United Kingdom, causing suboptimal health outcomes to the mother and child. Obesity and excessive gestational weight gain are risk factors for GDM. The Institute of Medicine recommends weight targets for women that are overweight and obese, however, there are no clear guidelines for women with GDM. Observational data suggest that modest weight loss (0.6–2 kg) after 28 weeks may reduce risk of caesarean section, large-for-gestational-age (LGA), and maternal postnatal glycaemia. This protocol for a multicentre randomised double-blind controlled trial aims to identify if a fully controlled reduced energy diet in GDM pregnancy improves infant birthweight and reduces maternal weight gain (primary outcomes). A total of 500 women with GDM (National Institute of Health and Care Excellence (NICE) 2015 criteria) and body mass index (BMI) ≥25 kg/m^2^ will be randomised to receive a standard (2000 kcal/day) or reduced energy (1200 kcal/day) diet box containing all meals and snacks from 28 weeks to delivery. Women and caregivers will be blinded to the allocations. Food diaries, continuous glucose monitoring, and anthropometry will measure dietary compliance, glucose levels, and weight changes. Women will receive standard antenatal GDM management (insulin/metformin) according to NICE guidelines. The secondary endpoints include caesarean section rates, LGA, and maternal postnatal glucose concentrations.

## 1. Introduction

Gestational diabetes (GDM) affects around 5% of pregnant women in the United Kingdom (UK) [1] and increases the risk of suboptimal materno-fetal outcomes and is associated with pre-pregnancy obesity and excessive gestational weight gain [2,3,4,5]. GDM is usually diagnosed at 24–28 weeks’ gestation and identifies women at risk of type 2 diabetes in later life. Although excessive gestational weight gain in early pregnancy (0–28 weeks) is well-established as a risk factor for GDM [3], the role of weight control in women after the diagnosis of GDM, from 28 weeks to delivery, is unclear.

Weight gain is a normal part of a healthy pregnancy, but excessive weight gain can contribute to poor outcomes for both the mother and child. Excessive gestational weight gain is currently defined using the Institute of Medicine guidelines (2009) based upon a woman’s pre-pregnancy body mass index (BMI) [6]. Normal weight, overweight, and obese women are advised to gain 11.4–15.9 kg, 6.8–11.4 kg, and 5.0–9.1 kg, respectively [6]. Many pregnant women exceed these targets and gain excessive weight [7]. It is also unclear if these targets are suitable for women with GDM, who are already at higher risk of adverse pregnancy outcomes compared with women without diabetes [5] and who may benefit from lower gestational weight gain targets [8]. It is unclear if maternal weight gain after GDM diagnosis remains important to prevent adverse maternal and baby outcomes, but there are data that suggest this is the case. A study by Harper and colleagues showed that for every 1 lb per week increase in weight in women after diagnosis with GDM, there was a 36–83% increased risk for pre-eclampsia, caesarean section, macrosomia, and large-for-gestational-age (LGA) (7). The guidelines of the National Institute of Health and Care Excellence (NICE) for diabetes in pregnancy highlight the importance of pre-pregnancy and post-partum weight control in women with GDM, but do not provide guidance on gestational weight gain targets [9]. There is currently limited evidence to guide clinical practice in this area.

Excessive weight gain has been associated with multiple adverse outcomes. In the general obstetric population, excessive gestational weight gain has been associated with hypertensive disorders in pregnancy [10], LGA [11], macrosomia [12], depression [13], and may also be linked to infant death [14]. In later life, women with excessive gestational weight gain during pregnancy are at increased risk of type 2 diabetes and cardiometabolic disease [15], which may be linked to the weight gained in pregnancy not being completely lost after the birth [16]. Offspring of women with excessive gestational weight gain have increased body weight, increased fat mass, and increased blood pressure in childhood, raising concerns about obesity and diabetes risk in later life [17,18,19].

Women with GDM and who are overweight or obese pre-pregnancy may benefit from minimal gestational weight gain in later pregnancy to improve both short and longer term health outcomes. Theoretically, reduced late gestational weight gain may improve infant birth weight and impact upon postpartum weight retention and future cardiometabolic risk. However, few studies have ever assessed pregnancy outcomes following moderate energy restriction in women with GDM [20,21]. These studies have been useful in providing evidence for maternal and neonatal outcomes, however they did not provide comprehensive evidence that having a strict food allocation over a set period of time could impact on these outcomes, as addressed in this protocol.

The clinical trial protocol presented here is a double blind, fully controlled dietary intervention using a novel approach of diet boxes. Diet boxes have become very popular commercially and contain all of an individual’s meals and snacks for the week delivered to their home or workplace. Meals are designed to be nutritionally balanced, healthy, appetising and require only minimal time and effort to cook at home. As participants can be randomised to receive a box providing reduced or standard energy diet, this overcomes many of the challenges faced in controlling and blinding nutritional studies in free-living pregnant volunteers.

The aim of this trial is to identify if a reduced energy diet in 500 pregnant women diagnosed with GDM can reduce gestational weight gain, improve pregnancy outcomes, and reduce postnatal glucose concentrations.

## 2. Trial Population and Methods

### 2.1. Trial Design and Ethical Approvals

DiGest (Dietary intervention in Gestational diabetes) is a multicentre fully controlled randomised double-blind dietary controlled trial. It involves recruiting 500 participants who have been diagnosed with GDM to take part in an eating energy controlled diet during the last 8–10 weeks of their pregnancy. The trial is estimated to take three years and is based in the East Anglia region. It is being conducted in accordance with the Declaration of Helsinki, and the protocol was approved by the Research Ethics Committee in Cambridge, UK (reference 18/WM/0191) and the National Health Service (NHS) Health Research Authority (IRAS 242924; ISRCTN 37866).

### 2.2. Participant Eligibility

Eligible pregnant women are identified following diagnosis of GDM at 20 to 30 + 6 weeks’ gestation using a standard clinical 75 g oral glucose tolerance test (OGTT) in accordance with the NICE glucose thresholds (fasting ≥5.6 mmol/L and ≥7.8 mmol/L at two hours) [9]. Eligible women require a pre-pregnancy BMI ≥25 kg/m^2^ (based on measured weight at booking, using validated scales), singleton pregnancy confirmed by ultrasound, and planned antenatal care at the same or another participating trial centre throughout pregnancy. Women will be recruited by research midwives, nurses, or by their physician/obstetrician. Study documentation will be provided in advance and women who wish to participate will be offered an appointment to discuss the study and consent to participate. All women will give their written informed consent before participation in any trial related activities.

### 2.3. The Following Exclusion Criteria Will Be Applied

Women with evidence of multiple pregnancy or severe congenital abnormality on ultrasound; termination planned; pre-existing comorbidities such as renal failure, severe liver disease, cardiac failure, and psychiatric conditions requiring in-patient admission; medications at the time of the OGTT that may interfere with results (e.g., high dose steroids, immunosuppressants); complications such as preterm labour, severe anaemia, or intrauterine growth restriction (IUGR) at GDM diagnosis; haemoglobin A1c (HbA1c) at diagnosis of GDM of >48 mmol/mol; previous diabetes diagnosis; specialised dietary requirement (e.g., vegan or severe nut allergy); and weight loss >5% based between pre-pregnancy weight (weight first measured in pregnancy at booking) and 28 weeks’ gestation.

### 2.4. Design of The Intervention

The diet boxes were developed in association with our industrial partner Mayfield Foods Ltd., Oxfordshire, UK. The diet boxes contain 2000 kcal/day (control) or reduced energy (1200 kcal/day) comprising 40% carbohydrate, 25% protein, and 35% fat. The macronutrient levels are similar between both diets, however, portion sizes in the control arm are significantly larger to account for the increase in calories.

Macronutrient composition was designed to be in line with current clinical practice for gestational diabetes to ensure the standard calorie diet box was comparable to a standard GDM diet. There is little clear evidence for dietary modification in GDM, but many centres recommend a slight reduction in carbohydrate [22] and slight increase in protein for satiety. Upon discussion with our trial steering committee, we agreed 40% carbohydrate and 25% protein, as higher protein consumption has been associated with reduced birthweight [23].

Carbohydrate sources were from low glycaemic index foods in line with the NICE guidelines [24]. The diet box meals do contain naturally occurring sugars, for example, in fruit, but they do not contain any added sugar, preservatives, or artificial flavour enhancers, and adhere to NHS food safety advice for pregnant women [25,26].

Participants order their food weekly from the DiGest study website. Each participant will receive seven breakfasts, lunches, and dinners, all supplied as frozen pre-prepared meals; seven snack-packs; and an additional optional pack containing raw vegetables and salads. As breakfast can often challenge blood glucose levels in women with GDM, both intervention and control groups have the same breakfast options (300 kcal), with the remaining daily dietary energy intake split between lunch, dinner, and snacks. To maximise choice, lunch and dinner options are interchangeable. Snacks are delivered in a daily snack-pack, which contains a similar mixture of carbohydrate, protein, and fat containing foods. Altogether, participants are provided with 10 breakfast options, 22 lunch/dinner options, and 8 snack-packs (Table 1). New menu items are added continuously and in response to feedback. There are sufficient options to cater for vegetarian and pescatarian women. Women are allowed to drink sugar-free, calorie-free drinks such as water, tea, or coffee with a teaspoon of milk, or sugar-free fizzy drinks. These are not provided in the diet box.

Lunch and dinner options contain 1–3 portions of vegetables per meal, and there are additional portions in the snack-pack and vegetable/salad packs. Initial reviews of the frozen meals, which were piloted on health care professionals (*n* = 16) and participants (*n* = 5), suggested that participants missed crunchy vegetables, and the vegetable pack was added to address this. Participants can choose between 4–9 portions of fruit and vegetables every day. Calcium requirements (1200 mg/day recommended by World Health Organisation (WHO) [27] are met using leafy green vegetables such as rocket, spinach, kale, wholemeal flour, milk, yogurt, cheese and cream. We did not supply extra calcium as this is not widely recommended in the United Kingdom and carries some risks [28,29]. However, many pregnant women also take multivitamins that contain a daily dose of 200–500 mg of calcium.

Feedback on the diet boxes will be received verbally from participants after the first diet box has been delivered. Women are also asked to complete a single questionnaire about the diet provided during the intervention. This feedback allows us to adjust the meal choices on offer, improve the diet box composition while still maintaining the macronutrient balance, and address any issues to ensure the diet boxes meet participants’ expectations.

### 2.5. Baseline Measurements and Randomisation

An outline of the protocol is detailed in Figure 1. Women who wish to participate and have given written informed consent are randomised before 30 + 6 weeks’ gestation.

#### 2.5.1. Clinical Measures

At recruitment, baseline blood pressure and anthropometric measurements (including weight, height, waist, mid upper arm circumference (MUAC) and neck circumferences, and skinfold thickness (SFT)) will be assessed. Participants will be measured barefoot and in light clothing. Weight will be recorded to the nearest 100 g using a calibrated electronic scale (Seca Hammer Steindamm). Height will be assessed to the nearest 0.1 cm with a calibrated wall-mounted stadiometer (SECA 286; Seca, Birmingham, UK). Body mass index (BMI; in kg/m^2^) will be calculated as weight divided by square height. The circumferences will be measured to the nearest 0.1 cm with a nonstretchable fiber-glass tape (Chasmors Ltd., London, UK). Waist circumference will be defined as the midpoint between the lowest rib margin and the iliac crest; MUAC as the mid-point between the tip of the shoulder and the tip of the elbow (olecranon process and the acromium); and the neck circumference as the midway of the neck, between the midcervical spine and midanterior neck. The SFT will be performed on the left side of the body from multiple sites, including triceps, biceps, suprailiac, and subscapular using a calibrated Harpenden skinfold caliper (Baty International, UK). Blood pressure will be taken once seated using a calibrated automatic oscillometric sphygmomanometer (Dinamap, machine) and systolic and diastolic blood pressure will be recorded. Measurements will be performed by trained research staff, following standard protocols. Quality control workshops will be set up during the trial, where study team members will conduct measurements alongside expert examiners to reinforce the importance of standardised procedures and to assess inter- and intra-observer errors.

#### 2.5.2. Laboratory Measures

Fasting blood will be taken by a nurse, research midwife, or trained phlebotomist at baseline, 36 weeks, and 6 weeks post-partum, processed within two hours, for laboratory analysis of markers of glucose metabolism (glucose, insulin, c-peptide, HbA1c), lipid profiles, and full blood count (if not already taken for clinical purposes). HbA1c is collected in an Ethylenediaminetetraacetic (EDTA) tube (Starstedt, Leicester, UK) and measured in local laboratories using a Tosoh High performance liquid chromatography (HPLC) analyser and aligned to the International Federation of Clinical Chemistry (IFCC) method. All laboratories participate in quality assurance and have acceptable performance. Serum and plasma form blood will be stored at −80 °C for future novel biomarkers of interest. Batch analysis of biomarkers will be performed by a centralized laboratory based at the Cambridge Biomedical Campus. A whole blood sample is collected in an EDTA tube (Starstedt, Leicester, UK) and stored at 4 °C for later DNA extraction. A lithium heparin tube (Starstedt, Leicester, UK) is used for insulin and C-peptide determination and is stored on ice immediately after withdrawal prior to centrifugation.

### 2.6. Dietary Assessment and Compliance

#### 2.6.1. Assessment of Normal Dietary/Energy Intake

Participants will be asked to complete questionnaires including the EuroQol Quality of life (EQ-5D-5L), three factor eating questionnaire 18 (TFEQ-18) [30], and online multiple pass 24 h dietary recalls (using the “Intake24” programme) [31]. Validity of total energy intake reported by Intake24 was compared to total energy expenditure using doubly labelled water in 98 U.K. adults [31]; the correlations were 0.31 (one recall), 0.47 (two repeated recalls), and 0.39 (three repeated recalls), respectively.

#### 2.6.2. Assessment of Adherence to the Diet Boxes

Women are asked to complete an Intake24 and a three-day food diary while wearing a masked continuous glucose monitor (CGM) for 7–10 days (Dexcom G6, San Diego USA), which is used for assessment of typical diet. Women are also asked to complete three-day food diaries at various points, while on the diet boxes, to assess adherence while wearing a CGM. Participants are asked to give specific details on all their food consumption including portion sizes and drinks. They are asked to ensure that one weekend day is included in this three-day monitoring period.

#### 2.6.3. Randomisation and Blinding

At the first visit, a unique participant number will be allocated by the recruitment team. Using the participant number, women will be randomly assigned to the energy restriction arm (1200 kcal/day) or the standard dietary arm (2000 kcal/day). This randomisation protocol was designed by a statistician based upon participant numbers and has been integrated within the online food ordering system. The allocation, therefore, is blinded from the participant, research team, and clinical teams, but is automatically available to the industrial partner for preparation of the diet box. This process is audited on a monthly basis to ensure the randomization is running smoothly for both arms of the study.

#### 2.6.4. Monitoring Period

Participants are advised to weigh themselves twice weekly on standard Bluetooth scales, provided on loan from the study team, to ensure that any weight loss is gradual and stable. They will have regular contact with the research team, which will be documented. Women will attend their standard antenatal appointments in which they will have weight checks, urinalysis, and ultrasound growth scans. Standard GDM care will be offered by clinical teams according to the NICE guidelines [9]. In brief, women are given diet and lifestyle advice after diagnosis and receive a glucometer with instructions on self-monitoring blood glucose. Medication (metformin and / or insulin) will be offered to women with glucose levels persistently over NICE targets (5.3 mmol/L fasting and 7.8 mmol/L at one hour) in line with the NICE guidance. Doses are up-titrated to achieve improved glycaemic control. Clinical teams making treatment decisions are unaware of the participant’s allocation [9].

### 2.7. Intervention Visits

Women are invited for a visit at 32 (visit 2) and 36 weeks (visit 3) gestation, using ±7 days for each visit. During visits 2 and 3, participants will be weighed, have blood pressure taken, a CGM sensor inserted, and asked to complete a three-day food diary while wearing the concurrent CGM during the next 7–10 days. At visit 2, women will be offered participant information leaflets about voluntary procedures around the time of the birth, specifically placental biopsy, cord blood collection, and infant anthropometry measures to be taken at birth and six weeks postnatal. These procedures will be discussed again at visit 3, when specific consent for these additional voluntary procedures will be sought. Visit 3 will include completing all the questionnaires, maternal blood measures, and maternal anthropometry performed in visit 1.

Visit 4 will take place six weeks postpartum. To avoid inconvenience to participants, this visit replaces the standard postnatal glucose test offered to all women after GDM [9]. At this visit, participants will have their anthropometry measurements repeated as per baseline and body composition measured by dual-energy X-ray absorptiometry (DXA) scan (Hologic, Manchester, UK) at sites where this facility is available. A urine sample will be taken for microalbuminuria. Participants will be asked to complete the EuroQol EQ5D and TFEQ-18 and provide brief details about infant feeding choices. A final CGM sensor will be applied and worn for 7–10 days. Neonatal anthropometry and air displacement plethsmography (ADP)-PEA pod (Cosmed Ltd., Bicester, UK) body composition measurements (where available) will be undertaken at this visit (subject to specific additional written consent). Anthropometry will be performed by trained paediatric research nurses or research assistants, following standard procedures. These measurements will include weight, length, MUAC, waist and head circumferences, and skinfold thickness (SFT). Weight will be measured to the nearest 1 g using a Seca 757 electronic baby scale (Seca, Birmingham, UK). Length will be assessed to the nearest 0.1 cm using an Infantometer (Seca 416, Birmingham, UK). Skinfold thickness will be performed in triplicate at four sites (triceps, subscapular, flank, quadriceps) on the left side of the body using a calibrated Holtain Tanner/Whitehouse Skinfold Caliper (Holtain, Crymych, UK). From ADP-PEA pod, body composition information including body volume, body density, body surface area, total fat mass (FM), total fat-free mass (FFM), and percentage body fat (%BF) will be derived.

### 2.8. Study Outcomes

There are two co-primary endpoints; maternal weight change between enrolment and 36 weeks’ gestation and neonatal birthweight. Neonatal sex-appropriate SD scores (SDS) will be calculated for weight and length measurements (with adjustments for gestational age at birth) using customised centiles (UK 1990 growth reference using LMS growth software LMSgrowth) [32].

#### Secondary Outcomes are Categorised for the Neonate and the Mother as Follows

Neonatal: Gestational age at delivery, preterm delivery (<37 weeks), large/small-for-gestational age (using local, national, and international centiles and customised centiles), amniotic fluid glucose, cord blood C-peptide, admission to the neonatal intensive care unit (NICU), duration of admission to NICU, neonatal jaundice requiring phototherapy, Apgar scores, body composition and anthropometry, neonatal hypoglycaemia (defined as a capillary glucose <2.6 mmol/L on one or more occasions, within the first 48 h of life starting at least 30 min after birth, and necessitating treatment either with 40% glucose gel administered to the buccal mucosa and/or with intravenous dextrose), neonatal nasogastric feeding, and feeding type on discharge from hospital. Infant feeding choices and feeding history will also be examined at six weeks postpartum. Neonatal anthropometry; for each of the four skinfold thickness measurements, an internal standard deviation score (SDS) will be calculated, using residuals from a linear regression model, adjusting for infancy age, gestational age at birth, and sex. Mean skinfold thickness SDS will be used in analyses. Birth ponderal index will be calculated by dividing the infant’s birthweight by its birth length cubed (kg/m^3^). Secondary analysis for sex-appropriate SDS scores will also be assessed using different growth centiles such as GROW centiles [33] and INTERGROWTH centiles [34].

Maternal outcomes at 32 weeks, 36 weeks and postpartum: Maternal outcomes to be studied include maternal weight, BMI, glycaemia (CGM metrics as per the international consensus recommendations [35], HbA1c), cardiometabolic health (blood pressure, lipids, fasting insulin, fasting glucose), maternal food choice and eating behaviour, quality of life, treatments administered for GDM, birth modality and complications, and postpartum glycaemia (measured by two-point OGTT and CGM).

### 2.9. Sample Size Justification

Retrospective data assessing the effect of late gestational weight gain (GWG) upon pregnancy outcomes were used to inform the sample size calculation [36]. The following sample size examples are based upon alpha 0.05 and power 0.9.

In the retrospective study, women had a standard deviation of 2 kg for late GWG and <3 kg for total GWG. Allowing for a standard deviation of 3 kg and a 1 kg difference between groups, 190 women per group will be required to give 90% power for the primary maternal endpoint.

The neonatal primary endpoint will be standardised birthweight. Recruitment of 175 women per group will give 90% power for identification of a 0.3 sd difference in standardised birthweight. As the standard deviation for birthweight was 508 g, this broadly equates to a difference in birthweight of 150 g.

A study size of 500 participants will provide sufficient statistical power and will allow for a 20–25% withdrawal rate. This sample size will also give sufficient power to detect secondary outcomes:To identify an increase in LGA (odds ratio (OR) 2.25 at 80% power).To identify a difference in maternal glycaemia at 36 weeks, for example, differences in mean CGM glucose (0.3 mmol/L difference), time in target (8% difference in time in target (3.5–7.8 mmol/L; >100 min per day), which has been associated with clinically relevant differences in neonatal outcomes) [8].To identify a 0.7 mmol/L difference in maternal postnatal two-hour OGTT glucose.

Several secondary analyses will be performed. A per protocol analysis will be performed in participants with >80% compliance and at least four weeks’ exposure to the intervention. Data will be used in a qualitative way to assess predictors for withdrawal or poor compliance. Consent will also be taken to collect and analyse outcomes for women who withdraw from the study, to allow assessment of reasons for withdrawal and their bearing upon outcomes.

### 2.10. Statistical Analysis

An intention to treat analysis of the co-primary outcomes will be based on t-tests between trial arms and linear regression models with adjustment for baseline variables and the stratification variable of study centre using a fixed effects model. In addition, the model for maternal gestational weight change will be adjusted for baseline weight as an explanatory variable. The potential role of other explanatory variables such as pre-pregnancy BMI or information from the questionnaires will be investigated in sensitivity analyses. Secondary outcomes will be examined through regression analyses (linear or logistic) appropriate for the type of outcome being considered. A further secondary analysis will be performed to assess if main results are altered with addition of early GWG to the regression model. Safety analyses will be performed to assess any quantitative difference in effects, rates of small-for-gestational-age (SGA)/IUGR, stillbirth, neonatal hypoglycaemia, and admission to NICU.

## 3. Discussion

This protocol describes a novel free-living fully controlled dietary energy restriction trial in pregnant women with GDM to assess the effect on maternal weight gain and neonatal birthweight. Using diet boxes to provide all meals and snacks to participants in both arms of the trial facilitates blinding and randomisation, in order to control dietary energy intake insofar as possible. It also ensures that dietary intake is in line with guidelines for a safe and healthy diet during pregnancy.

The objective of this study is to assess the impact of maternal dietary energy restriction on maternal and neonate outcomes. Previous attempts to define clear targets for gestational weight gain in women with GDM have been hindered by two main concerns: firstly, that calorie restriction in pregnancy may be harmful, and secondly, that intervening after 28 weeks is too late. Data from the Dutch Famine Winter cohort have demonstrated that women exposed to severe calorie restriction during pregnancy (<500–800 kcal/day) have infants of lower birth weight who are at higher risk of SGA and IUGR [37]. However, in clinical and research populations with more modest energy restriction, there has been no evidence of harmful effects. For example, women with hyperemesis gravidarum give birth to infants with no evidence of growth restriction [38]. Studies of modest energy restriction in GDM have demonstrated no adverse effects [20,21], with SGA rates comparable to the baseline population [20,21]. A recent study demonstrated that a reduced energy diet in pregnant women with GDM is acceptable to patients and associated with a good quality of life [39]. Hodson and colleagues assessed the effects of a low calorie diet in 16 women with GDM on liver triacylglycerol content [39]. Women were recruited at 21–34 weeks’ gestation and followed a four-week diet supplying 1200 kcal/day (50% carbohydrate, 25% protein, 25% fat). The diet was well tolerated and women lost a mean of 0.4 ± 0.4 kg per week during the four-week intervention. The study was not randomised and did not aim primarily to assess pregnancy outcomes [39]. This study also demonstrated that calorie restriction in pregnancy is achievable in women with GDM and was well tolerated. The study used a research dietician to give one-to-one support to each participant using a meal plan. Although this was very successful, this delivery mode for the intervention placed a high burden on participants and required substantial staff resources.

Interventions to change the overall energy intake of the diet in free-living conditions are challenging, with many studies having design limitations that reduce the validity of the results (reviewed in [40]). For example, randomisation and blinding in free-living trials is challenging and often not performed. Nutritional studies may be entirely self-reported and portion sizes cannot be controlled [41,42]. The diet boxes may support participants in following the prescribed diet by providing all the meals and snacks in an easy to follow format. The portions are controlled and all the nutrition required for a safe and healthy diet during pregnancy is provided.

Controlling gestational weight gain after 28 weeks may only give a relatively short time for intervening, but previous work has suggested that this still could have beneficial effects upon pregnancy outcomes [21]. Although there are no long term data about dietary interventions after 28 weeks, overall, gestational weight gain has an impact upon women’s weight for up to 15 years after the pregnancy [16]. There are several studies that support the potential benefits of controlling gestational weight gain after a diagnosis of GDM, and that address the issues of safety and timescales. We recently completed a retrospective study of 547 women with GDM [36]. Overall total gestational weight gain (0–36 weeks) was associated with increased rates of LGA, caesarean section, and reduced rates of vaginal deliveries. Late GWG was also associated with postpartum glucose homeostasis with a positive association with two-hour OGTT glucose concentration on the postpartum OGTT, raising the possibility that controlling GWG for a short 8–10 week period could lead to longer-term beneficial effects upon glucose tolerance and diabetes incidence [36].

DiGest is a randomised controlled blinded trial assessing whether an energy restricted diet in pregnant women with GDM can improve pregnancy outcomes for mother and child. Although the current evidence base suggests that controlling pregnancy-related weight gain can improve health outcomes, much of the traditional advice given to women in pregnancy today with growing rates of overweight and obesity is inappropriate. The common myth of ‘eating for two’ may be particularly detrimental for women with GDM.

## Figures and Tables

**Figure 1 nutrients-12-01165-f001:**
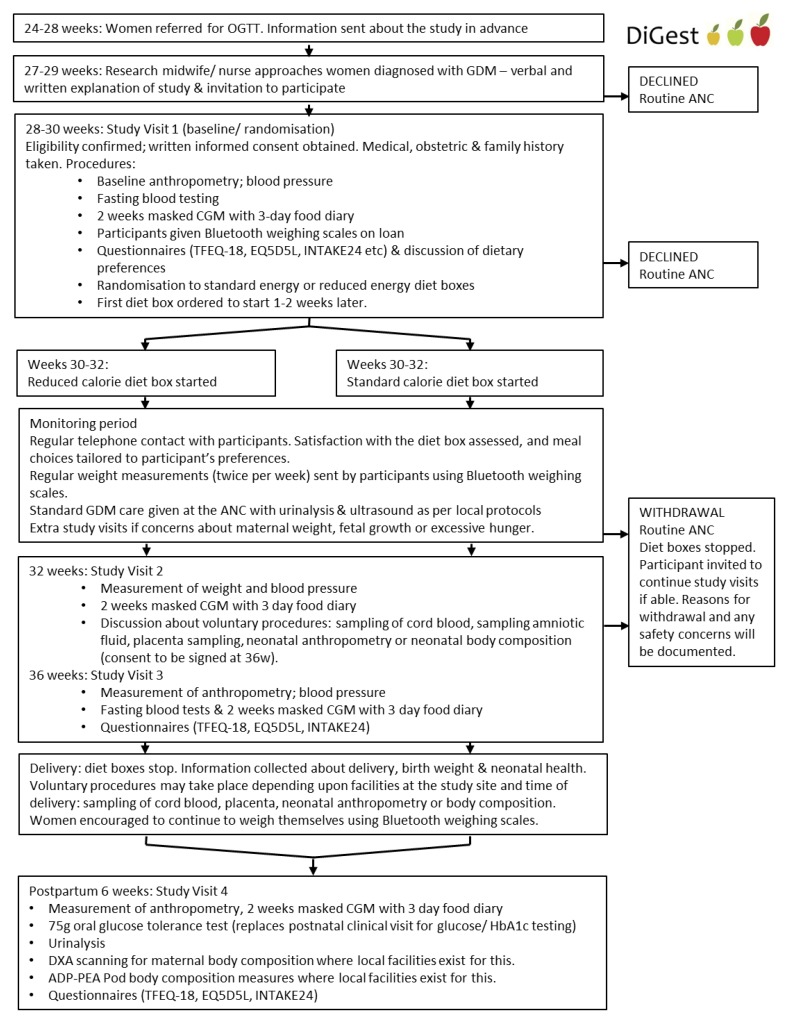
DiGest (Dietary intervention in Gestational diabetes) protocol summary. Abbreviations: OGTT, oral glucose tolerance test; GDM, gestational diabetes mellitus; ANC, antenatal clinic; CGM, continuous glucose monitoring; TFEQ–18, three factor eating questionnaire 18; EQ5DL, EuroQol 5 dimension quality of life questionnaire; HbA1c, hemoglobin A1c; INTAKE24, dietary recall questionnaire over 24 h period; DXA, dual-energy X-ray absorptiometry scan; ADP–PEA, air displacement plethysmography pea pod.

**Table 1 nutrients-12-01165-t001:** Example food options for a week on the study. Each day will come with an additional snack-pack and there is also a salad/veg box provided if requested.

Weekday	Breakfast	Lunch	Dinner	Snack Pack
**Monday**	Porridge with nuts and jam	Chilli bean wrap	Turkey roast	Boiled egg, satsuma, small cheese
**Tuesday**	Cheese and ham omelette with rosti	Mushroom stroganoff with rice	Macaroni cheese with kale	Apple, Belgian Chocolate covered rice cake, spiced seeds
**Wednesday**	Breakfast roll	Chicken schnitzel, wedges, and green beans	Venison sausage in red wine sauce with sprouts	Cottage cheese, Ryvita, satsuma
**Thursday**	Spiced omelette with Sag Aloo	Seafood lasagne	Beef madras with rice	Peperami (mini), orange, Belgian chocolate rice cake
**Friday**	Blueberry yogurt	Edamame and feta wrap	Salmon with lemon Puy lentils	Peperami (mini), pear, spiced seeds, popcorn
**Saturday**	Granola	Spiced Moroccan chicken wrap	Fish goujon, wedges, and minted peas	Small cheese, apple, Philadelphia snack light herbs, and breadsticks
**Sunday**	Cheese and mushroom omelette	Thai red chicken curry with rice	Vegetarian bean stew, rice, and halloumi	Satsuma, Belgian chocolate covered rice cake, spiced seeds
**Weekly Veg/Salad Pack**	Contains a range of vegetables and salad options including carrots, broccoli, cauliflower, baby tomatoes, cucumber, lettuce, celery, and red pepper			

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
