# Peer review of "Dietary Intervention in Pregnant Women with Gestational Diabetes; Protocol for the DiGest Randomised Controlled Trial"

_nutrients, 2020, doi:10.3390/nu12041165_

Round 1

Reviewer 1 Report

The authors present a protocol of a study assessing a fully controlled reduced energy diet in GDM pregnancy in a randomized controlled setting. The study design is novel and well described. There are no ethical concerns. The possible low birth weight/IUGR issue is well addressed. The sample size calculations are well justified and include both primary and other outcomes. 

There are only minor comments, that concern Figure 1: not all abbreviations are explained (INTAKE24, DXA, ADP-PEA), additionally it is unclear what "amniotic fluid" in the figure refers to?

Reviewer 2 Report

This paper describes the rationale and design of the DiGest (Dietary Intervention in pregnant women with gestational diabetes mellitus (GDM)) randomized clinical trial.   Women with GDM diagnosed after 28 weeks gestation are randomised to standard or reduced energy diet.  Primary outcome measures are maternal weight gain and birth weight centile.

Suggestions for changes to text:

Line 95: Section 2.2 Participant eligibility

More information is required regarding pre-analytical OGTT protocol across study sites e.g. is fluoride-oxalate or fluoride-citrate-EDTA tube used for measurement of plasma glucose NB tube type is very important as it can have a large impact on diagnosis of GDM.

Line 98: pre-pregnancy BMI >25 kg/m2

                Is this self-reported or calculated from weight at booking?

Line 111: HbA1c at baseline visit of >48 mmol/mol.

I presume HbA1c is used to identify and exclude women with undiagnosed pre-existing diabetes.  This requires clarification.  Does baseline visit refer to booking antenatal visit or the study visit 1.  If it is the former, perhaps call it booking visit.  If it is the latter, this is the first reference in the paper to the study baseline visit.  This requires clarification in the text and needs to be added to the flow chart, e.g. excluded participants coming off the Study Visit 1 box.

Line 112: weight loss >5% pre-pregnancy weight between booking and 28 weeks gestation

Is this self-reported pre-pregnancy weight or weight at booking.  If weight at booking, please refer to it as booking weight.

Line 143: Initial reviews of the frozen meals…..

If the study is already underway, describe the numbers e.g. initial reviews in (n) participants suggested….

Line 184: Section 2.5.2

This section is very important.  Please provide more details, e.g.:

  • Whole blood collected into EDTA tube (company, city, country) for HbA1c
  • Whole blood collected into fluoride-oxalate or fluoride-citrate-EDTA tube (company, city, country) for measurement of plasma glucose NB glucose instability due to glycolysis can have a particularly large impact on fasting samples
  • Is a centralized laboratory used for study investigations or local labs?

Line 188: Blood will be stored…

Is an aliquot of whole blood stored or is serum or plasma separated, aliquoted and stored at -80C?  Or both?

Line 216: Standard GDM case will be offered by clinical teams….

What does Standard GDM case mean?

Line 221: Women are invited for a visit at 32 (visit 2) and 36 weeks (visit 3) gestation…

Considering some women may not commence the diet box until week 32, is there also a window of two weeks for these visits?

Line 222: …have blood pressure taken once whilst seated (using calibrated electronic Dinamap machine

Is this consistent with the method reported in Section 2.5.1 for BP measurement? If so, add these details in section 2.5.1 and do not repeat here.

Line 233: Dual energy X-ray absorptiometry (DXA) scan

Please cite company, city & country.  NB dual does not require capitalization and there should be a hyphen between dual and energy.

Line 236: ADP-PEA Pod body composition measurements….

                Please cite company, city & country

Line 247: Section 2.8 Study outcomes

Please explain why GROW centiles are not used as the primary endpoint and only used as the secondary analysis.  Is ethnicity recorded as part of baseline history, if so GROW centiles will take different ethncities into account.

Suggestions for changes flow diagram:

Some of the information in the flow diagram is inconsistent with the text:

Box 3: 28-30 weeks: Study Visit 1 (baseline/randomization)

Urinalysis is listed as a procedure however the only details in the text are found in Line 215 and Line 234 (NB also be consistent with use of terminology urine-analysis v urinalysis).  What urine analytes are being measured for study purposes and how are they measured?

Line 198, section 2.6.2 Assessment of adherence to the diet boxes

Clarification is required.  The text suggests CGM is only worn once women have been randomised to the diet boxes, however the flow chart shows that women also complete 2 weeks masked CGM with 3-day food diary prior to randomization.

Box 5: Monitoring period

Regular weight measurement (1 per week) in box is not consistent with Line 212 which reads participants are advised to weigh themselves twice weekly

Box 6: 32 weeks: Study Visit 2 & 36 weeks: Study Visit 3

Line 228: Visit 3 will include completing all the questionnaires….

This suggests that the questionnaires are not completed in Visit 2, however this is not consistent with the flow diagram.

Formatting and use of acronyms:

Please use space before units of measurement and be consistent with defining acronyms.  Some examples are:

Line 36: Gestational diabetes mellitus (GDM) - be consistent with use of acronym eg line 121

Line 38: gestational weight gain (GWG) - be consistent with use of acronym throughout paper

Line 36: United Kingdom (UK)

Line 45: 15.9 kg

Line 97: 75 g oral glucose tolerance test (OGTT) – define OGTT

Line 146: World Health Organization (WHO)

Line 161: OGTT; oral glucose tolerance test

Line 164: HbA1c; haemoglobin A1c

Line 167: mid upper arm circumference (MUAC)

Line 206: 1200 kcal/day

Line 255: Neonatal intensive care unit – neonatal does not require capitalization

Line 268: Section 2.8.2 - start a new paragraph under this heading as used for 2.8.1

Line 307: define small for gestational age (SGA) – be consistent with use of acronym in discussion

Line 350: Caesarean - does not require capitalisation

Line 373: References have double numbering

Reviewer 3 Report

I had already read the manuscript and overall I would like to make my compliments for the authors for their study protocol. I did not discover major flaws that need to be changed.

Author Response

We thank the reviewer for their kind comments.